# Size Matters in the Cytotoxicity of Polydopamine Nanoparticles in Different Types of Tumors

**DOI:** 10.3390/cancers11111679

**Published:** 2019-10-29

**Authors:** Celia Nieto, Milena A. Vega, Jesús Enrique, Gema Marcelo, Eva M. Martín del Valle

**Affiliations:** 1Departamento de Ingeniería Química y Textil, Facultad de Ciencias Químicas, Universidad de Salamanca, 37008 Salamanca, Spain; celianieto@usal.es (C.N.); mvega@usal.es (M.A.V.); j.enrique@usal.es (J.E.); 2Departamento de Química Analítica, Química Física e Ingeniería Química, Facultad de Farmacia, Universidad de Alcalá, 28801 Alcalá de Henares (Madrid), Spain

**Keywords:** polydopamine nanoparticles, size, cytotoxicity, iron affinity, doxorubicin

## Abstract

Polydopamine has acquired great relevance in the field of nanomedicine due to its physicochemical properties. Previously, it has been reported that nanoparticles synthetized from this polymer are able to decrease the viability of breast and colon tumor cells. In addition, it is well known that the size of therapeutic particles plays an essential role in their effect. As a consequence, the influence of this parameter on the cytotoxicity of polydopamine nanoparticles was studied in this work. For this purpose, polydopamine nanoparticles with three different diameters (115, 200 and 420 nm) were synthetized and characterized. Their effect on the viability of distinct sorts of human carcinomas (breast, colon, liver and lung) and stromal cells was investigated, as well as the possible mechanisms that could be responsible for such cytotoxicity. Moreover, polydopamine nanoparticles were also loaded with doxorubicin and the therapeutic action of the resulting nanosystem was analyzed. As a result, it was demonstrated that a smaller nanoparticle size is related to a more enhanced antiproliferative activity, which may be a consequence of polydopamine’s affinity for iron ions. Smaller nanoparticles would be able to adsorb more lysosomal Fe^3+^ and, when they are loaded with doxorubicin, a synergistic effect can be achieved.

## 1. Introduction

Nanomedicine has acquired an essential role in recent decades due to the urgent need for developing novel therapeutic strategies for cancer treatment. This multidisciplinary field, by taking advantage of nanotechnology, aims to overcome the pharmaceutical limitations of conventional chemotherapeutics. Thus, the improvement of their stability, aqueous solubility and dose limiting toxicity, as well as avoiding the apparition of multidrug resistances (MDR), are some of the objectives that are highly relevant for the scientific community today [1,2]. Moreover, with the employment of nanoparticles (NPs) for cancer treatment, increasing tumor retention of therapeutic molecules while reducing their non-specific distribution in normal tissues is being attempted [2].

In this manner, a considerable number of nanomaterials have been developed recently [3,4,5]. Among them, polydopamine (PD), a synthetic melanin, which is a black biopolymer produced by autoxidation of dopamine, has achieved great importance [6]. Thanks to their physical and chemical properties [5], their feasible surface functionalization, and their good degradation in vivo [7,8], PD NPs stand out for the development of new diagnosis and antiproliferative nanosystems [5,9,10,11,12,13,14]. 

Thus, there are only a few existing works that report nanoparticle systems devoid of drugs with an antineoplastic behavior in the literature [15,16]. However, in previous studies, it was demonstrated that PD NPs can also constitute themselves an antitumour strategy [17,18]. Thereby, unlike the majority of developed therapeutic nanocarriers, their use could help to reduce the administration of drugs. In one of these previous works, it was found that PD by itself had an antiproliferative capacity after developing a novel protocol to evaluate its cytotoxicity [18]. Furthermore, it was reported that such antiproliferative capacity could be related to a ferroptosis mechanism, since PD cytotoxicity was reduced in the presence of an iron chelator (deferoxamine, DFO) or a potent antioxidant compound involved in cellular ROS-protection (glutathione, GSH) [18,19,20]. As with other melanins [17,21], PD presents a great binding affinity for free metallic ions, especially for Cu^2+^ and Fe^3+^ [17,22,23]. These free ions, which are more abundant in cancer cells, are stored in lysosomes and, since PD NPs end up in these organelles when they are internalized [18], their metal affinity could be related to their cytotoxicity [24,25,26,27,28]. Precisely because of this fact, in another previous study, the antiproliferative ability of iron-loaded PD NPs was investigated, and it was observed that such systems had a strong cytotoxicity for breast carcinoma cells [17].

On the other hand, it is well accepted for the design of therapeutic nanosystems that size may determine their endocytosis rate and, therefore, their antiproliferative activity [29,30,31]. Cancer endothelium is known to present leaky fenestrations between vasculature cells, and these gaps have sizes ranging from 100 to 780 nm, depending on the tumor type. As a consequence, it is generally considered that the smaller the size of the administered NPs, the higher their rate of endocytosis and their accumulation in tumor sites [30,31,32].

For all these reasons, the size-dependent cytotoxicity effect of PD NPs was studied in the present work. PD NPs with three different sizes (115, 200 and 420 nm) were synthetized and characterized. Their antiproliferative activity was analyzed and compared in four distinct types of human carcinomas (breast, colon, liver and lung carcinomas) and in healthy stromal cells. Moreover, it was again proven that such antiproliferative effects may be related to their metal-loading ability. To do that, their cytotoxicity was studied in co-treatment experiments of PD with either DFO (iron chelator) and GSH (antioxidant compound), conducted with all of the previous mentioned cell lines [32,33,34]. Finally, doxorubicin (DOX), a conventional chemotherapy drug [35,36], was adsorbed on PD NPs in a very low concentration (10.6 ng DOX/mg PD NP) with the goal of enhancing their antitumor impact while reducing the necessary DOX therapeutic doses. The antitumor effect of the resulting nanosystem over the viability of breast carcinoma cells was also studied.

## 2. Results

### 2.1. Synthesis and Characterization of 115, 200 and 420 nm PD NPs

PD NPs of different sizes were synthetized through dopamine oxidative polymerization in a basic aqueous medium [17,18,37]. It contained fixed volumes of ethanol and water and different concentrations of ammonium hydroxide, on which the size of PD NPs depends (Figure 1a). To characterize them, transmission electron microscopy (TEM) images of all PD NPs were taken and size-range histograms were obtained (Figure 1b–d). As shown, their average sizes were 75 nm, 170 nm and 340 nm. Thus, the higher the ammonium hydroxide concentration employed, the smaller the synthetized PD NPs. In trizma base solution (50 mM, pH 10), such NPs remained stable with average hydrodynamic diameters of 115 ± 50 nm (PdI = 0.047), 199 ± 50 nm (PdI = 0.085) and 417 ± 50 nm (PdI = 0.012). Such values were analyzed by dynamic light scattering (DLS) and were higher than the ones obtained by TEM because of a possible PD NPs hydration.

### 2.2. Cytotoxicity Effect of PD NPs Depends on Their Size

Since it was demonstrated that size of NPs plays an important role in their endocytosis rate and therapeutic effect [29,30,31], and that PD NPs are able to reduce the viability of cancer cells themselves [17,18], it was analyzed how the variation of PD NPs’ diameter could affect to their cytotoxicity in different types of tumors. In this manner, BT474, HTC116, HEPG2 and H460 human cell lines were selected to carry out MTT assays with previously synthetized 115 and 200 nm PD NPs (Figure 2a–d and Appendix A) [38]. Furthermore, such assays were also performed with a stromal human cell line, the HS5 one, in order to elucidate whether there were significant variations in the viability reduction of treated cancer and healthy cells (Figure 2e and Appendix A). Four different concentrations of PD NPs were employed (from 0.0074 mg/mL to 0.042 mg/mL), and MTT assays were performed following a protocol which was previously developed in order to avoid the strong contribution of PD NPs absorbance to formazan salts absorbance values [18].

As shown in Figure 2a and Appendix A, when breast cancer cells (B7474 cell line) were treated with more elevated PD NPs concentrations, a more noticeable reduction in their viability was achieved. Thus, the best cytotoxic effect was obtained 72 h after treatment of the mentioned cells with 0.042 mg/mL of 100 nm PD NPs, which reduced their survival rate to 59%. The same concentration of 200 nm PD NPs decreased BT474 viability to 65% after 72 h.

In the case of the treatment of colon and liver carcinoma cells (HTC116 and HEPG2 cell lines) (Figure 2b,c and Appendix A), it was analogously found that the best cytotoxic effect was accomplished when 0.042 mg/mL of 100 nm PD NPs were employed. With these experimental conditions, HTC116 survival rate was reduced to 63% after 72 h and HEPG2 viability, to 60%.

Likewise, a similar trend in the results obtained after the treatment of lung carcinoma cells (H460 cell line) can be observed in Figure 2d and Appendix A. Nevertheless, the cytotoxicity of PD NPs was less remarkable than for the three previously mentioned tumor cell lines (BT474, HTC116 and HEPG2). Thereby, 72 h after the treatment with 0.042 mg/mL of 100 nm PD NPs, H460 survival rate decreased to 74%.

Positively, in the same manner, stromal cells’ viability was not reduced as much as the carcinoma cell lines’ viabilities when they were treated with PD NPs, neither those PD NPs with a size of 100 nm nor those with a size of 200 nm (Figure 2e and Appendix A). Treatment with the highest concentration (0.042 mg/mL) of such NPs reduced, after 72 h, the viability of HS5 cells to 75–76%, but this value was higher than 80% 72 h after the treatment with the lowest concentrations of 100 and 200 nm PD NPs.

Therefore, in brief, it was found that the cytotoxicity of PD NPs was more noticeable, at all administered concentrations, for the HTC116, HEPG2 and, especially, for the BT474 cell line. The lowest concentration (0.0074 mg/mL) was barely toxic during the studied time for all treated cell lines. However, the two highest concentrations (0.029 and 0.042 mg/mL) achieved a notable decrease in cancer cell viability 72 h after a treatment of 35–40%, except for H460 viability, which was not so remarkably reduced. Furthermore, PD NPs did not decrease the viability of stromal cells in the same manner as the viability of cancer cells. In fact, the two lowest concentrations employed (0.0074 and 0.015 mg/mL) of PD NPs showed pretty specific results. Treatment with them almost had no affect on HS5 cellular viability, possibly because tumor cells present high intracellular iron levels [25,26]. Finally, it was generally observed that the smaller the diameter of the PD NPs, the more pronounced their cytotoxic effect. In comparison with the antiproliferative effect of the 200 nm PD NPs, those with a 115 nm diameter achieved a decrease in the viability of BT474 with a 3–8% improvement in efficacy, depending on the administered concentration. These differences in viability reduction, in the case of the HTC166, HEPG2, H460 and HS5 cells, were close to 2–9%, 3–12%, 2–5% and 1–11%, respectively, and could be explained by a larger endocytosis rate.

Next, since it was corroborated that size affected the therapeutic effect of PD NPs, MTT assays were again performed with synthetized 420 nm PD NPs [18]. For such assays, the BT474 cell line was chosen because the best previous MTT results were obtained with these breast carcinoma cells. As a result, and as shown in Figure 3, after the first 24 h of treatment, none of the concentrations employed were cytotoxic. The three lowest concentrations had virtually no effect on BT474 survival rate, and the highest one (0.042 mg/mL) only achieved a reduction in their viability to 92%. After 48 h of treatment, the cytotoxicity of the 420 nm PD NPs, in all concentrations, was neither pretty noticeable but, when 72 h had elapsed, it was more remarkable for the three highest chosen concentrations. In this manner, 0.015 mg/mL of PD NPs reduced breast cancer cells’ survival to 84%, which was then decreased to 79% and 75% when 0.029 mg/mL and 0.042 mg/mL of PD NPs were employed for the treatment of BT474 cells. In any case, as expected, the cytotoxicity of 420 nm PD NPs was lower than that caused by the 115 and 200 nm PD NPs, with there being a difference of as high as 20–25% in cellular viability when the two most elevated NPs concentrations (0.029 and 0.042 mg/mL) were employed.

Finally, in order to visually corroborate the cellular viability results obtained in the MTT assays, a live/dead assay with confocal laser scanning microscopy (CLSM) was performed. For this purpose, BT474 cells were again chosen and treated for 72 h with 0.042 mg/mL of 115 nm PD NPs, which were the NPs and the concentration that had previously reduced the survival of the breast carcinoma cell line the furthest. 24, 48 and 72 h after treatment, these cells were stained with calcein AM and propidium iodide (PI), and the resulting images are shown below in Figure 4 [39,40,41]. As can be observed, the survival rate of BT474 cells was reduced to 74% after the first 24 h of treatment and, after 48 and 72 h, 57% and 48% of the breast cancer cells remained alive, with these survival rates being similar to those that had been obtained when MTT assays were carried out previously (Figure 2a and Appendix A). Thus, images of the live/dead assay corroborate the MTT results, wherein 115 nm PD nanoparticle cytotoxicity was more remarkable after 48 h of treatment.

### 2.3. PD NPs Cytotoxicity in Tumor Cells Could be Related to Their Iron Affinity

PD NPs’ ability to adsorb Fe^3+^ has been demonstrated in previous studies, as well as the fact that when these NPs are endocyted [17,42], they end up in cellular lysosomes. Such organelles are specifically responsible for the regulation of the concentration of free metallic cations and, as a consequence, when cells are treated with PD NPs, ROS production levels may be increased due to a possible imbalance in Fenton chemistry [18,24] (Figure 5).

For this reason, ferroptosis could be the process of cell death responsible for the cytotoxicity of PD NPs [32,34]. To demonstrate this fact, more MTT assays were performed with the same cell lines (BT474, HTC116, HEPG2, H460 and HS5) [17], treating cells with 115 and 200 nm PD NPs (0.029 mg/mL), but also with non-toxic concentrations of DFO (0.7 μM) and GSH (50 μM), an iron chelator and an antioxidant compound, respectively [19,20] (Figure 6a).

The results obtained with the BT474 and the HS5 cell lines when 115 nm PD NPs were employed are shown below (Figure 6b,c). It can be observed that, in both cell lines, the co-treatment with DFO and with GSH was able to decrease the antiproliferative effect of PD NPs at all measured times in all cell lines, with this fact becoming more noticeable after 24 h of treatment. For this reason, when BT474 cells were co-treated with DFO or GSH and with PD NPs, these last ones were around 12–18% less successful, depending on the time at which they were measured. In the case of the HS5 cell line, the cytotoxicity of PD NPs was reduced by 5–11%. This reduction could be lower because PD NPs were not as toxic for this cell line as for BT474 cells; however, despite this, with the administration of DFO and GSH, the toxicity of PD NPs was almost totally reduced for HS5 cells. Ultimately, with the other tumor cell lines (Appendix A) and with a co-treatment of DFO (0.7 μM) or GSH (50 μM) with 200 nm PD NPs (0.028 mg/mL), similar results as the ones obtained after the co-treatment of BT474 cells were also observed.

### 2.4. DOX-Adsorbed PD NPs (DOX@PD NPs) Presented a Notable Antiproliferation Activity

DOX is one of the most widely employed antineoplastic agents for the treatment of different types of cancers today, but its administration is currently limited due to the severe adverse effects that it has for patients. In this manner, many strategies are being developed to encapsulate this drug in order to improve its side toxicity [43,44]. Among these, PD-modified DOX-nanocarriers can already be found in the literature [45,46]. This issue, together with the fact that DOX is related to the formation of iron-related free radicals in cells [35,36], propitiated its selection for loading on the PD NPs synthetized here. For this purpose, 115 nm PD NPs were chosen because, as has been shown (Figure 2 and Appendix A), they present a more remarkable antiproliferative activity. In this manner, these NPs were mixed with a diluted DOX solution (10 nM), and the concentration of DOX adsorbed on PD NPs was determined by difference, by measuring the absorbance (λ = 494 nm) of the filtered supernatant once the DOX@PD NPs were isolated. As a result, it was found that the PD NPs were able to be loaded with 10.6 ng DOX/mg PD NPs.

Based on this data, three distinct concentrations of DOX@PD NPs (0.029, 0.035 and 0.045 mg/mL) were tested in BT474 cells, which were again chosen because the best MTT assay results had been achieved with this breast carcinoma cell line. Likewise, treatment with equivalent DOX concentrations (0.5, 0.7 and 0.8 nM), which were pretty low in comparison to those employed in other works [11,44], was performed.

The results of these experiments indicated that, as shown in Figure 7, after only 24 h of treatment, DOX@PD NPs presented noticeable cytotoxicity, especially at the two highest concentrations (0.035 mg/mL + 0.7 nM and 0.042 mg/mL + 0.8 nM). While PD NPs achieved reductions in BT474 survival rate to 76%, 74% and 68%, the same concentrations of DOX@PD NPs achieved reductions of 73%, 62% and 50%, respectively. Moreover, the increase in the cytotoxicity of the two highest concentrations was more potent after 48 and 72 h of treatment. On the one hand, after 48 h had elapsed, 0.035 and 0.042 mg/mL of DOX@PD NPs decreased breast carcinoma cell viability to around 52% and 44%, while viability was approximately 72% and 64% when the same concentrations of non-loaded PD NPs were employed. On the other hand, after 72 h of treatment with the mentioned concentrations of DOX@PD NPs, BT474 viability was around 42% and 37%, while the corresponding values for similar treatment with PD NPs were 65% and 58%. In this way, at all measured times, cellular viability reduction caused by DOX@PD NPs was more remarkable than that caused by the non-loaded PD NPs, chiefly when cells were treated with 0.035 and 0.042 mg/mL of NPs. Furthermore, DOX@PD NPs were even more effective than the equivalent treatment with DOX concentrations, especially during the first 24 h. After this time, the highest concentration of free DOX only achieved a reduction in BT474 survival to 91%, and the other two presented almost no cytotoxicity. Then, 48 and 72 h after treatment, the reduction in cellular viability caused by free DOX was more noticeable but, in any case, it did not exceed that resulting from equivalent treatment with DOX@PD NPs. Such results could be explained by a likely more remarkable ferroptosis-dependent ROS production that could take place together with DOX-dependent DNA damage [35,36]. In this manner, the use of DOX@PD NPs could enhance the therapeutic activity of both PD and DOX, resulting in lower DOX doses being required for potential treatment, partially avoiding the severe adverse effects that this drug has for patients, as mentioned above [43,44].

## 3. Discussion

Since nanomedicine’s occupation of a fundamental role in the development of novel therapeutic systems, the scientific community has been aware of the importance possessed by size in determining their effect [29,30,31,47]. For this reason, in the present work, the influence of size on the cytotoxicity of PD NPs, previously reported for breast and colon carcinoma cells [17,18], was studied in different tumor and stromal cell lines.

Thus, PD NPs with three different diameters (115, 200 and 420 nm) were synthetized. Once they had been characterized by TEM and DLS, the cytotoxicity of four different concentrations of the two smallest types of PD NPs (115 and 200 nm) was studied in different types (breast, colon, liver and lung) of human carcinomas and in stromal cells. Furthermore, the largest PD NPs’ (420 nm) effect on cellular viability was also tested with the breast carcinoma cell line, BT474. The results obtained showed that, in all cases, the smaller the size of the NPs, the more noticeable their antiproliferative activity, with this being even more noticeable in BT474 cells. In addition, with such experiments, it could be noticed that the lowest tested concentrations of PD NPs had a tumor-selective effect. In this manner, employing low doses of these NPs could be a possible strategy for achieving greater treatment specificity and overcoming the any undesirable side effects. So far, most developed therapeutic nanosystems are vehicles for the delivery of cytotoxic drugs and, in this way, the employment of PD NPs could represent an awesome strategy for reducing the administration of chemotherapeutics [15]. Likewise, the size of PD NPs could be selected according to the antiproliferative effect that would be desired.

On the other hand, it was demonstrated that the adsorption of iron could be responsible for the cytotoxic effect of such PD NPs. Their affinity for different metallic cations has been described in numerous studies, and the ability of PD NPs to load lysosomal Fe^3+^ cations once they have been endocyted could explain the death of cells treated with them by means of Fenton chemistry [17,18,24,34]. To corroborate that this ferroptosis process could take place in the present study, more viability assays were carried out, treating cells of the different carcinomas simultaneously with PD NPs and with two iron chelators, DFO and GSH. In this manner, it was demonstrated that this co-treatment affected the cytotoxicity of PD NPs, decreasing it at all measured times and in all types of treated cells.

Finally, 115 nm PD NPs were loaded with DOX. This drug was chosen because it is one of the most commonly used chemotherapeutic drugs, and because it has also been associated with ferroptosis processes in recent toxicity-related studies [35,36]. The cytotoxicity of the resulting NPs (DOX@PD NPs’) was analyzed with BT474 cells and compared with the viability values obtained with equivalent concentrations of free-DOX and non-loaded PD NPs. On the basis of this comparison, it could be seen that the antiproliferative effect of DOX@PD NPs was considerably greater. Actually, these NPs were even more effective than free DOX, possibly because of the occurrence of a synergist effect. Additionally, the DOX doses employed were much lower than those used so far in literature, and this fact could be important in avoiding the side effects, too [12,20].

## 4. Materials and Methods

### 4.1. Chemicals

Dopamine Hydrochoride, Ammonium Hydroxide (NH_4_OH), Phosphate Buffered Saline (PBS, 0.01 M, pH 7.4), Dulbecco’s Modified Eagle’s Medium (DMEM), Fetal Bovine Serum (FBS, USA origin), Thiazolyl Blue Tetrazolium Bromide (MTT reagent), L-glutathione Reduced (GSH) and Deferoxamine Mesilate CRS (DFO) were all supplied by Sigma-Aldrich (Darmstadt, Germany). Ethanol absolute was purchased from VWR International Eurolab S.L. (Llinars del Vallés, Barcelona, Spain). Penicillin-Streptomycin (5000 U/mL), Calcein AM, Propidium Iodide (PI) ReadyProbes^TM^ reagent and Doxorubicin Hydrochloride Solid (DOX) were obtained from Thermo Fisher Scientific (Eugene, OR, USA).

### 4.2. Synthesis and Characterization of PD NPs of Different Sizes

PD NPs with average diameters of 115, 200 and 420 nm were synthetized by mixing distinct volumes of NH_4_OH aqueous solution (3.9, 2 and 1.1 mL, respectively, 28–30%) with ethanol (40 mL) and deionized water (90 mL). In all cases, the obtained mixture was left under magnetic stirring at room temperature for 30 min, and a dopamine hydrochloride aqueous solution (10 mL, 50 mg/mL) was later added. The reaction was allowed to react for 24 h. PD NPs were isolated and purified by several centrifugation-redispersion cycles in deionized water (35 mL). Finally, NPs were re-suspended in PBS at final concentrations of 2, 3.2 and 1 mg/mL, respectively [17,18,35].

To characterize them, TEM images (Tecnai Spirit Twin, Fei Company, Hillsboro, OR, USA) were taken, employing a voltage acceleration of 120 kV. The synthetized PD NPs were dispersed in deionized water (pH = 6.0) in a concentration of less than 0.01% (WT), and a drop of this dispersion was deposited on a copper grid with a collodium membrane and dried for 24 h. Images were recorded after this time, and histograms exhibiting the size ranges were obtained. To this end, the size of a minimum of 300 PD NPs in different images was determined (using ImageJ software, NIH, Bethesda, MD, USA) in order to build each histogram. Moreover, the hydrodynamic diameter of the PD NPs was analyzed using DLS (on the basis of their intensity–average size distribution) (Zetasizer Nano ZS90, Malvern Instruments Inc., Royston, Hertfordshire, UK) when they were re-suspended in a trizma base solution (pH = 10.0) in a concentration that was also less than 0.01% (WT).

### 4.3. Cell Culture

BT474, HTC116, HEPG2, H460 and HS5 cell lines were cultured with medium supplemented with FBS (10%) and antibiotics (penicillin-streptomycin) (1%) as instructed (ATCC, Wesel, Germany). All cells were cultured at 37 °C in a humidified atmosphere in the presence of CO_2_ (5%).

### 4.4. Size-Dependent Cytotoxicity Effect of PD NPs

The antitumor activity of 115 and 200 nm PD NPs was firstly studied based on MTT assays [37]. BT474, HTC116, HEPG2, H460 and HS5 cells were seeded in 24-well plates and cultured with supplemented medium overnight. The next day, the culture medium was replaced by supplemented medium containing PBS (control) and four different concentrations (from 0.0074 mg/mL to 0.042 mg/mL) of 115 and 200 PD NPs. In addition, BT474 cells were also treated with the same concentrations of 420 nm PD NPs. In all cases, cellular survival was analyzed for 4 days (EZ Reader 2000, Biochrom, Cambridge, UK), every 24 h, following the protocol described by Nieto et al. [18]. The results shown are the mean ± SD of three replicates for each different treatment.

Secondly, the reduction of viability caused by 115 nm PD NPs was corroborated in BT474 cells by CLSM (Leica TCS SP5, Leica Microsystems, L’Hospitalet de Llobregat, Spain) with an alive/dead cellular assay [36]. The cells were seeded in crystal-bottom plates (8000 cells/mL) and cultured with supplemented medium overnight. Over the following days, the culture medium was replaced by supplemented medium containing PBS (control) and 115 nm PD NPs (0.042 mg/mL). After 24, 48 and 72 h, calcein AM (1.25 mM) and PI (1 drop/1 mL culture media) were added and, after 30 min, CLSM images were taken [37,38].

### 4.5. PD Iron-Affinity could Explain the Cytotoxicity of PD NPs in Tumor Cells

To demonstrate that the iron affinity of PD could be implicated in the cytotoxicity of cells treated with PD NPs, HTC116, HEPG2, H460, BT474 and HS5 cells were seeded in 24-well plates and cultured with supplemented medium overnight. After 24 h, the culture medium was replaced by supplemented medium containing PBS (control), DFO (0.7 μM), GSH (50 μM), 115 and 200 nm PD NPs (0.029 mg/mL) and PD NPs plus DFO and GSH at the same concentrations. Again, cell viability was analyzed every 24 h for 4 days, following the same protocol as before [18]. The results shown are also the mean ± SD of three replicates for each different treatment.

### 4.6. DOX Adsorption onto PD NPs

With the aim of adsorbing DOX onto PD NPs, an aqueous solution (1% DMSO) of the drug was prepared at a final concentration of 10 nM. PD NPs (0.75 mL, 0.035 mg/mL) with a diameter of 115 nm were mixed with DOX solution, and the mixture was kept under stirring overnight [17]. Next, DOX@PD NPs were isolated through centrifugation and re-suspended in PBS at a final concentration of 2 mg/mL. The resulting supernatant was preserved, and its DOX concentration was determined in order to quantify PD NPs DOX adsorption. To this end, the supernatant was filtered several times with 0.1 µM syringe filters (GE Healthcare Life Sciences, Buckinghamshire, UK) and its absorbance was measured at 494 nm (UV-1800 Spectrophotometer, Shimadzu Corporation, Soraku-gun, Kyoto, Japan). Finally, taking into account the absorbance value of the initial DOX solution at such wavelength, the DOX adsorption onto PD NPs was quantified.

### 4.7. Antiproliferative Activity of DOX@PD NPs

To study the reduction in tumor cells’ viability caused by DOX@PD NPs, the BT474 cell line was chosen to carry out further MTT assays. Following the steps described above, the cells were again seeded in 24-well plates and, 24 h later, they were treated with medium supplemented with PBS (control), different concentrations of 115 nm DOX@PD NPs (0.029, 0.035 and 0.042 mg/mL) and equivalent adsorbed DOX concentrations (0.5, 0.7 and 0.8 nM). Cellular viability was measured every day for 72 h, always following the same protocol as before [18]. The results shown are again the mean ± SD of three replicates for each different treatment.

### 4.8. Statistical Analysis

Data related to the size of the PD NPs is the mean ± SD of three different measurements. Otherwise, results of the different MTT assays are represented as the mean ± SD of three replicates for each treatment of three different experiments, and the results were considered statistically significant where *p* < 0.05.

## 5. Conclusions

In this work, it has been demonstrated that size plays an important role in the cytotoxic effect of PD NPs. This was demonstrated on the basis of viability assays carried out with different human carcinoma cell lines and with stroma cells, in which the therapeutic action of two or three PD NPs of different sizes was analyzed (115, 200 and 420 nm). In all cell lines, and with all of the tested PD NP concentrations, it was shown that the smaller the diameter of the NPs, the more enhanced their antiproliferative activity. Therefore, the size of PD NPs could be tailored depending on the desired application.

With respect to the cytotoxic effect of PD NPs, it was observed that this may be related to a process of ferroptosis. PD NPs are able to load Fe^3+^ in lysosomes, and this fact could originate an abnormal cellular ROS production. However, when either an iron chelator (DFO) or a potent antioxidant (GSH) were employed, their antiproliferative ability was notably decreased. For this reason, when PD NPs were loaded with DOX, their antiproliferative effect was enhanced. Apart from triggering DNA damage, DOX cytotoxicity is also a consequence of a process of ferroptosis which, in addition to the one potentially produced by PD NPs, may result in the achievement of a synergist effect.

## Figures and Tables

**Figure 1 cancers-11-01679-f001:**
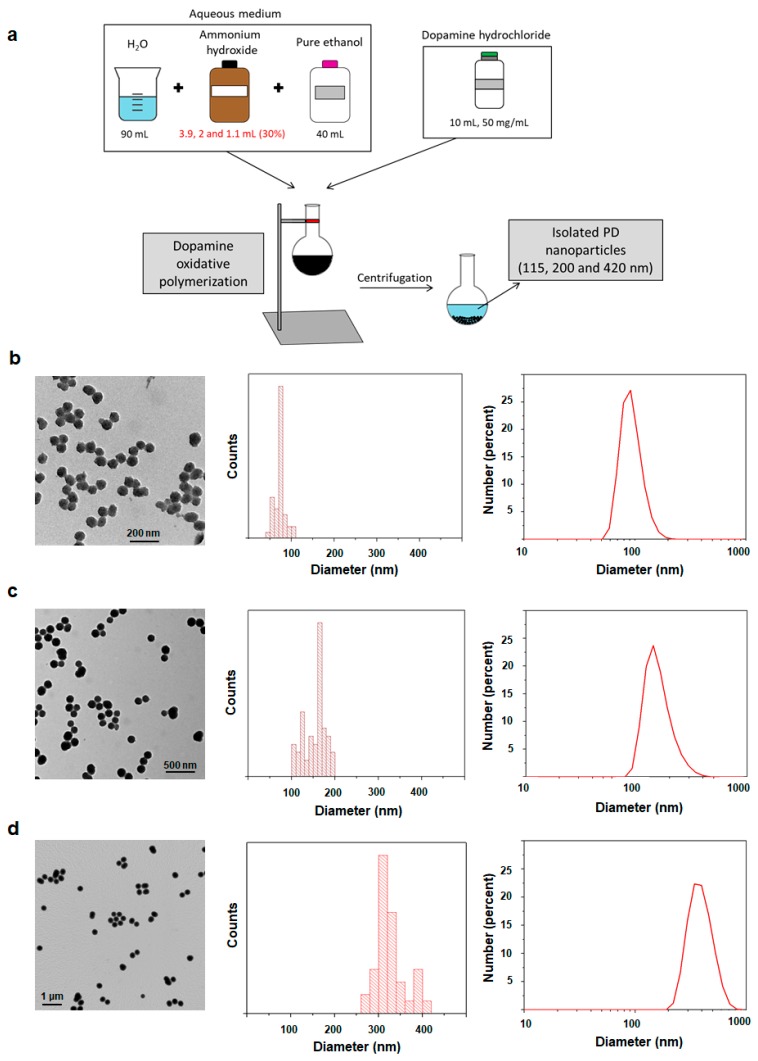
(**a**) Schematic representation of PD NPs synthesis by dopamine oxidative polymerization in a basic aqueous medium. (**b**–**d**) TEM images, size-range histograms and DLS number distributions of dispersions of PD NPs with three different diameters (pH = 7.0).

**Figure 2 cancers-11-01679-f002:**
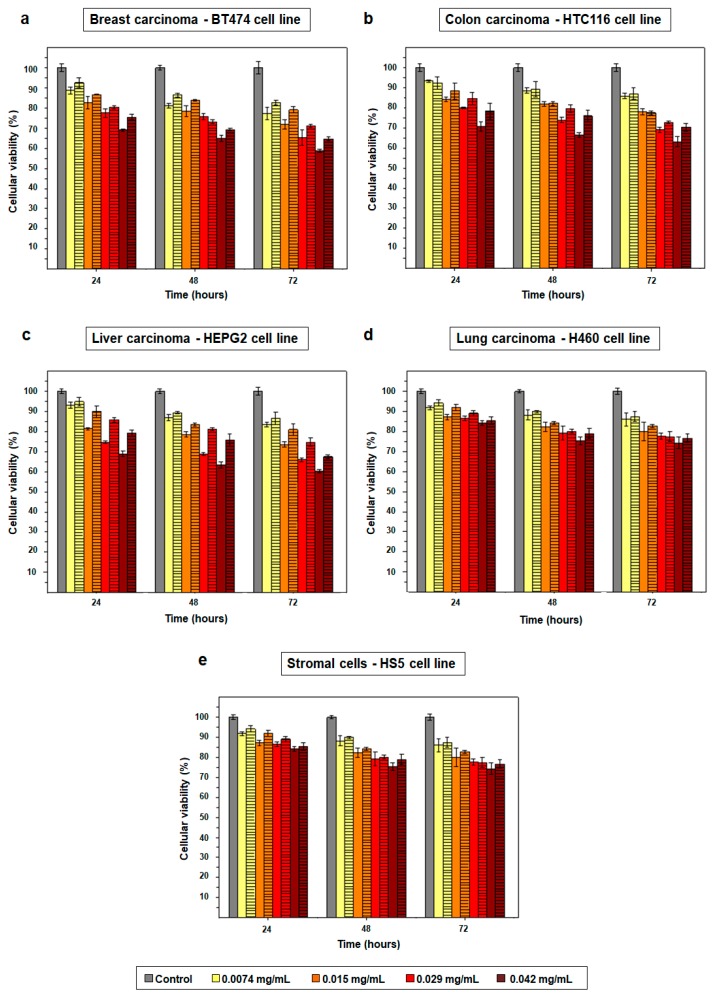
Obtained results in the MTT assays performed with the BT474 (**a**), HTC116 (**b**), HEPG2 (**c**), H460 (**d**) and HS5 (**e**) cell lines. Cells were treated with four different concentrations (0.0074, 0.015, 0.029 and 0.042 mg/mL) of PD NPs with a diameter of 115 nm (empty bars) and 200 nm (bars with the horizontal line pattern). The results shown are the mean ± SD of three replicates for each treatment.

**Figure 3 cancers-11-01679-f003:**
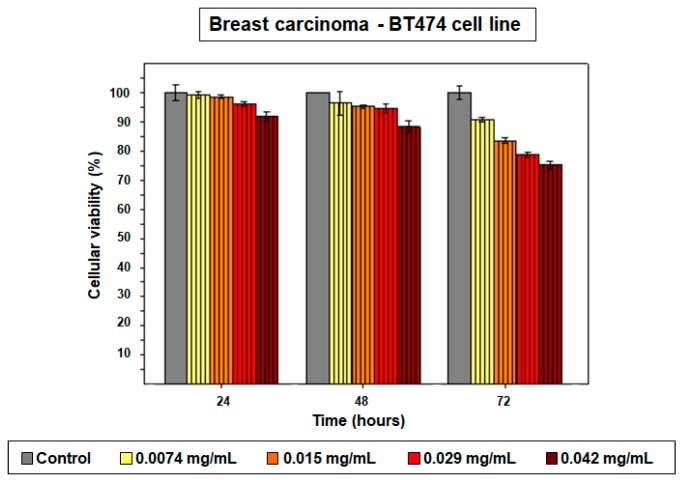
The results of the MTT assay carried out with PD NPs with a size of 420 nm (vertical lines—pattern bars) with BT474 cells. The same four concentrations (0.0074, 0.015, 0.029 and 0.042 mg/mL) of PD NPs as before were employed. The results shown are the mean ± SD of three replicates for each treatment.

**Figure 4 cancers-11-01679-f004:**
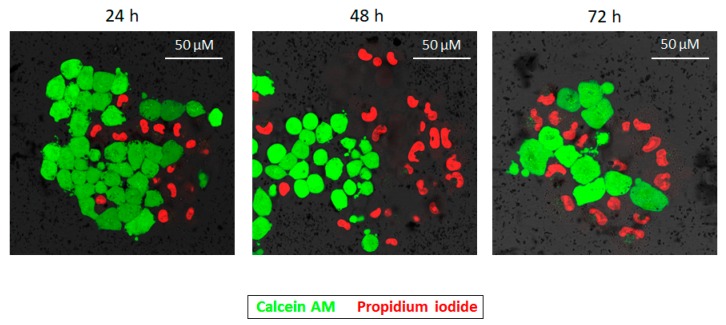
CMLS images taken 24, 48 and 72 h after the treatment of BT474 cells with 115 nm PD NPs (0.042 mg/mL). Calcein AM (in green, excited at 495 nm, indicator of cellular viability) and PI (in red, excited at 493 nm, indicator of cellular death) were selected to be performed in an alive/dead assay.

**Figure 5 cancers-11-01679-f005:**
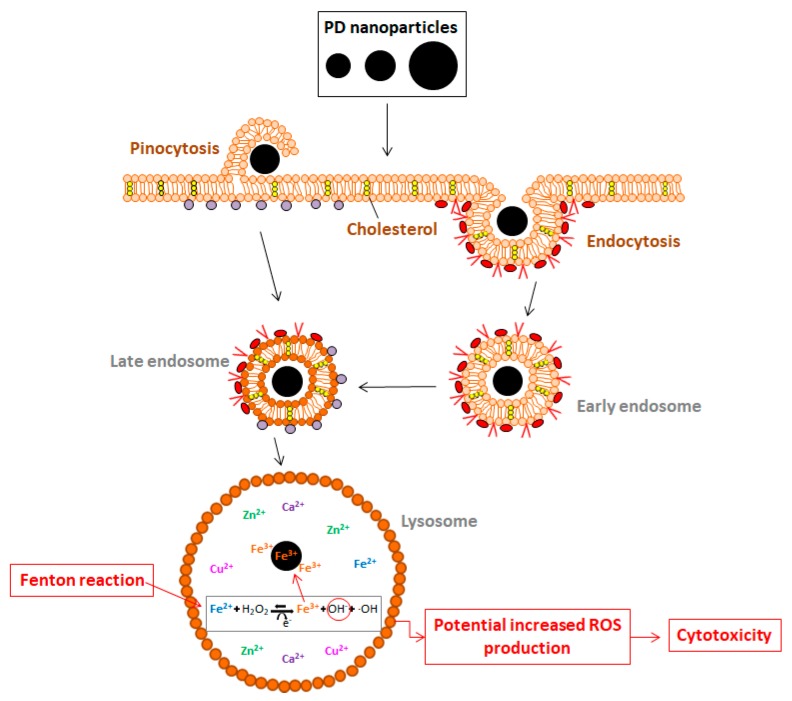
Schematic representation of the mechanism that could explain how PD NPs could be responsible for a reduction in cancer cells’ viability after their cellular internalization in lysosomes.

**Figure 6 cancers-11-01679-f006:**
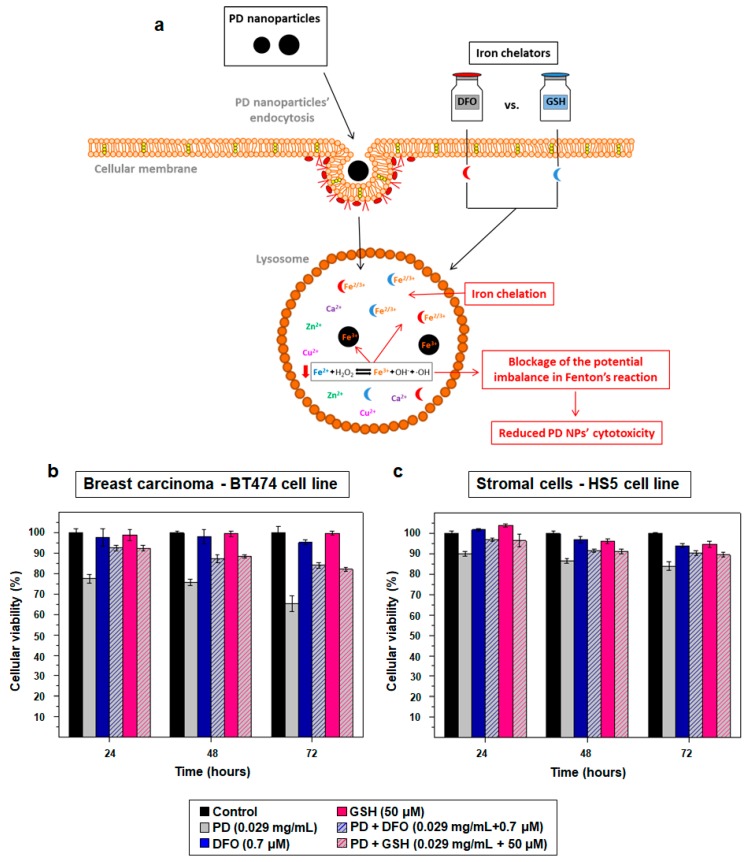
(**a**) Scheme that represents how DFO and GSH could be able to reduce the cellular death caused by the treatment with PD NPs. (**b**,**c**) Results of the MTT assays carried out with BT474 and HS5 cells after a co-treatment of 115 nm PD NPs (0.029 mg/mL, gray) with DFO (0.7 μM, blue) and GSH (50 μM, pink). The results shown are again the mean ± SD of three replicates of each treatment.

**Figure 7 cancers-11-01679-f007:**
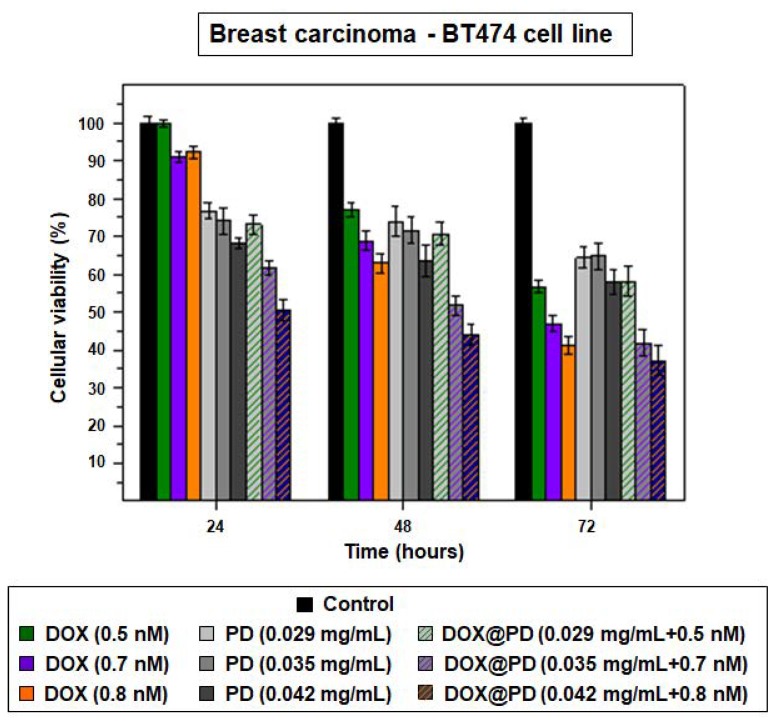
Results of the MTT assay carried out with the BT474 cell line, previously treated with free DOX (0.5 (green), 0.7 (purple) and 0.8 (orange) nM), 115 nm PD NPs (0.029, 0.035 and 0.042 mg/mL (grayscale)) and DOX@PD NPs (grayscale with colored patterns) in equivalent concentrations. The results shown are the mean ± SD of three replicates for each treatment.

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
