# Peer review of "Size Matters in the Cytotoxicity of Polydopamine Nanoparticles in Different Types of Tumors"

_cancers, 2019, doi:10.3390/cancers11111679_

Round 1
Reviewer 1 Report
Polydopamine has become a worldwide referent in many different areas nowadays, from functional coatings to Smart textiles. In this sense the authors report an innovative work on a new application of this material for the antiproliferative activity of anticancer cells. The authors had already described this effect though in this work they perform a more detail study about the nanoparticle size effect on the antitumoral activity. The work is well developed and the conclusions are of interest for the community, so I would recommend its publication.
However, before so, the uahtors should mkake an extra effort to demonstrate the mechanism described in figure 5. It is not clear at all how the chelation of iron by PD can favour ROS mechanism, as long as the uso of external chelators as GSH is able to decrease the antiproliferative effect of NPs. Please describe in more detail the whole mechanism, preferably with additional experiments that support this conclusion.
Author Response
Please see the attachment PDF file.

Reviewer 2 Report
This work submitted by Valle reported the synthesis of polydopamine (PDA) nanoparticles with three different size (115, 200, and 420 nm) and their cytotoxicity for several cancer cells and stromal cells. The ferroptosis process was therefore validated and proposed for PDA-based systems regarding their cytotoxicity. Furthermore, the authors showed that doxorubicin-loaded PDA nanoparticles (115 nm) can improve therapeutic efficacy due to the synergistic effect. The results were well-organized and this manuscript presented a clear story. It can help the readers in this field to have a better understanding of the cytotoxicity from PDA. I would like to recommend its publication in Cancers after addressing the following concerns.
In this work, smaller nanoparticles showed higher cytotoxicity. The authors ascribe this to their higher rate of endocytosis. However, as a key information presented in this manuscript, I didn’t see any direct experimental evidence. Could the authors provide some data to support this statement? In Figure 4, some black dots were observed in the background of the overlaid images. They seem some aggregation to me. Therefore, the colloidal stability of these PDA nanoparticles in diverse mediums should be also considered. DLS or other techniques were recommended. The sentence “PD NPs with a larger size could be selected if non-toxic NPs were searched for other applications (like diagnosis), since it was demonstrated that PD NPs of a 420 nm diameter present a much lower cytotoxicity (line 269)” is not accurate. The term “non-toxic” should have not been used here, since larger PDA nanoparticles also showed a certain level of toxicity. Relevant important work relating to PDA for cancer therapy should be also cited properly in this manuscript. For example, ACS Nano, 2018, 12, 2643-2651. In the scheme of Figure 1, “Dopamine chlorohydrate” should be changed to “Dopamine hydrochloride”.
Reviewer 3 Report
NietoNieto et al. described the influence of polydopamine nanoparticles' sizes on their cytotoxicity in different cell lines.
The manuscript is interestiong but the following points need to be addressed before its publication.
The reference 3 concerns mainly polydopamine NPs. the authors have to give a more general reference on nanomaterials Page 2, line 79, TEM: the authors have to give more details on the conditions used to record the TEM images: how are prepared the samples? What was the accelerating voltage? Page 2, line 80, "size-range histograms": how did the authors obtain such size-range histograms? Page 2, line 82; DLS analysis in trizma base solution": why don't the authors realize DLS analysis on NPs solutions analyzed by TEM? in this case also, the authors have to give more details on their DLS analysis (methods used, NPs concentration, etc.). Page 3, figure 1: the quality of the Figure 1A has to be improved. indeed, it is difficult/impossible to read what is written on the bottles of the aqueous media. Moreover, the curves for the DLS number distributions of dispersions of PD NPs are almost invisible. The authors have also to indicate the scale on their TEM images. The way the authors determine the encapsulated Dox is not really clear. Moreover, the initial absorbance of Dox solution was measured with 1% DMSO aqueous solution while the absorbance of the was supernantant measured in an aqueous solution without DMSO. Normally, a calibration curve has to be realized with the Dox solubilized in the same solvent than the one used for analyzing the drug loading.
